# CausalBench: A Large-scale Benchmark for Network Inference from Single-cell Perturbation Data

## Abstract

Mapping biological mechanisms in cellular systems is a fundamental step in early-stage drug discovery that serves to generate hypotheses on what disease-relevant molecular targets may effectively be modulated by pharmacological interventions. With the advent of high-throughput methods for measuring single-cell gene expression under genetic perturbations, we now have effective means for generating evidence for causal gene-gene interactions at scale. However, inferring graphical networks of the size typically encountered in real-world gene-gene interaction networks is difficult in terms of both achieving and evaluating faithfulness to the true underlying causal graph. Moreover, standardised benchmarks for comparing methods for causal discovery in perturbational single-cell data do not yet exist. Here, we introduce CausalBench - a comprehensive benchmark suite for evaluating network inference methods on large-scale perturbational single-cell gene expression data. CausalBench introduces several biologically meaningful performance metrics and operates on two large, curated and openly available benchmark data sets for evaluating methods on the inference of gene regulatory networks from single-cell data generated under perturbations. With real-world datasets consisting of over 200 000 training samples under interventions, CausalBench could potentially help facilitate advances in causal network inference by providing what is - to the best of our knowledge - the largest openly available test bed for causal discovery from real-world perturbation data to date.

## 1 Introduction

Studying causality in real-world environments is often challenging because uncovering causal relationships generally either requires the ability to intervene and observe outcomes under both interventional and control conditions, or a reliance on strong and untestable assumptions that can not be verified from observational data alone (Stone (1993); Pearl (2009); Schwab et al. (2020); Peters et al. (2017)). In biology, a domain characterised by enormous complexity of the systems studied, establishing causality frequently involves experimentation in controlled in-vitro lab conditions using appropriate technologies to observe response to intervention, such as for example high-content microscopy (Bray et al. (2016)) and multivariate omics measurements (Bock et al. (2016)). High-throughput single-cell methods for observing whole transcriptomics measurements in individual cells under genetic perturbations (Dixit et al. (2016); Datlinger et al. (2017; 2021)) has recently emerged as a promising technology that could theoretically support performing causal inference in cellular systems at the scale of thousands of perturbations per experiment, and therefore holds enormous promise in potentially enabling researchers to uncover the intricate wiring diagrams of cellular biology (Yu et al. (2004); Chai et al. (2014); Akers & Murali (2021); Hu et al. (2020)).

However, while the combination of single-cell perturbational experiments with machine learning holds great promise for causal discovery, making effective use of such datasets is to date still a challenging endeavour due to the general paucity of real-world data under interventions, and the difficulty of establishing causal ground truth datasets to evaluate and compare graphical network inference methods (Neal et al. (2020); Shimoni et al. (2018); Parikh et al. (2022)). In order to progress the causal machine learning field beyond reductionist (semi-)synthetic experiments towards potential utility in impactful real-world applications, it is imperative that the causal machine learning

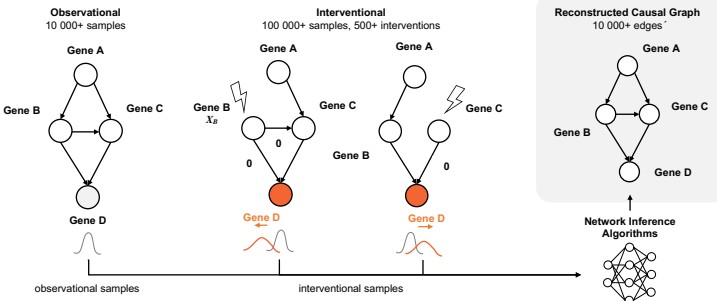

Figure 1: An overview of causal gene-gene network inference in mixed observational and perturbational single-cell data. The causal generative process in its unperturbed form is observed in the observational data (left; 10 000+ samples in CausalBench) while data under genetic interventions (e.g., CRISPR knockouts) are observed in the interventional data (right; 200 000+ samples in CausalBench. Either observational or interventional plus observational data that were sampled from the true causal generative process (bottom distributions) can be used by network inference algorithms (bottom right) to infer a reconstructed causal graph (top right) that should as closely as possible recapitulate the original underlying functional gene-gene interactions.

research community develop and maintain suitable benchmarks for objectively comparing methods that aim to advance the causal interpretation of real-world interventional datasets.

To facilitate the advancement of machine learning methods in this challenging domain, we introduce CausalBench - a comprehensive benchmark suite for evaluating network inference methods on perturbational single-cell RNA sequencing data that is - to the best of our knowledge - the largest openly available test bed for causal discovery from real-world perturbation data to date (Figure 1). CausalBench contains meaningful biologically-motivated performance metrics, a curated set of two large-scale perturbational single-cell RNA sequencing experiments with over 200 000 interventional samples each that are openly available, and integrates numerous baseline implementations of state-of-the-art methods for network inference from single-cell perturbational data. Similar to benchmarks in other domains, e.g. ImageNet in computer vision (Deng et al. (2009)), we hope CausalBench can help accelerate progress on large-scale real-world causal graph inference, and that the methods developed against CausalBench could eventually lead to new therapeutics and a deeper understanding of human health through enabling the reconstruction of the functional gene-gene interactome. The source code for CausalBench is openly available at https://github.com/ananymous-43213123/causalbench.

**Contributions.** Our contributions are as follows:

- We introduce CausalBench - a comprehensive benchmark suite for evaluating network inference methods on perturbational single-cell RNA sequencing data consisting of two curated, openly available benchmark datasets with over 200 000 interventional samples each. We introduce a set of meaningful benchmark metrics for evaluating performance including a novel statistical metric that leverages single cell perturbational data to measure performance against a larger set of putative gene regulatory relationships than would be possible using observational data alone.
- Using CausalBench, we conduct a comprehensive experimental evaluation of the performance of state-of-the-art network inference algorithms in recovering graphical relationships for mixed observational and interventional scRNAseq data. We implement relevant state-of-the-art methods as baselines for network inference from observational and interventional single-cell data.
- In addition, we evaluate the performance and scaling characteristics of network inference methods under varying numbers of available training samples and intervention set sizes to establish whether state-of-the-art network inference algorithms are able to effective use of different scales of intervention and training sample set sizes.

## 2 RELATED WORK

**Background.** Given an observational data distribution, several different causal networks or directed acyclic graphs (DAG) could be shown to have generated the data. The causal networks that could equally represent the generative process of an observational data distribution are collectively referred to as the Markov Equivalence Class (MEC) of that DAG (Huang et al. (2018)). Interventional data offers an important tool for limiting the size of the MEC to improve the identifiability of the true underlying causal DAG. In the case of gene expression data, modern gene editing tools such as CRISPR offer a powerful mechanism for performing interventional experiments at scale by altering the expression of specific genes for hundreds of genes at once and observing the resulting interventional distribution across the entire transcriptome (Dixit et al. (2016); Datlinger et al. (2017; 2021)). The ability to leverage such interventional experiment data at scale could significantly improve our ability to uncover underlying causal functional relationships between genes and thereby strengthen our quantitative understanding of biology. Establishing causal links between genes can help implicate genes in biological processes causally involved in disease states and thereby open up new opportunities for therapeutic development (Mehrjou et al. (2022); Shifrut et al. (2018)).

**Network inference in mixed observational and interventional data.** Learning network structure from both observational and interventional data presents significant potential in reducing the search space over all possible causal graphs. Traditionally this network inference problem has been solved using discrete methods such as permutation-based approaches (Wang et al. (2017); Hauser & Bühlmann (2012). Recently, several new models have been proposed that can differentiably learn causal structure (Schölkopf et al. (2021)). However, most of them focus on observational datasets alone. Ke et al. (2019) presented the first differentiable causal learning approach using both observational and interventional data. Lopez et al. (2022) improved the scalability of differentiable causal network discovery for large, high-dimensional datasets by using factor graphs to restrict the search space, and Scherrer et al. (2021) introduced an active learning strategy for selecting interventions to optimize differentiable graph inference.

**Gene-regulatory network inference.** The problem of gene regulatory network inference has been studied extensively in the bioinformatics literature in the case of observational datasets. Early work modeled this problem using a Bayesian network trained on bulk gene expression data (Friedman et al. (2000)). Subsequent papers approached this as a feature ranking problem where machine learning methods such as linear regression (Kamimoto et al. (2020)) or random forests (Huynh-Thu et al. (2010); Aibar et al. (2017)) are used to predict the expression of any one gene using the expression of all other genes. However, Pratapa et al. (2020) showed that most gene regulatory network inference methods for observational data perform quite poorly when applied to single-cell datasets due to the large size and noisiness of the data. In the case of constructing networks using interventional data, there is relatively much lesser work given the recent development of this experimental technology. Dixit et al. (2016) were the first to apply network inference methods to single-cell interventional datasets using linear regression.

In contrast to existing works, we exhaustively test several different network inference approaches for this task using a combination of both observational and interventional data. We present the first standardized benchmark for single-cell causal inference that enables a direct comparison of state-of-the-art machine learning methods using a set of quantitative metrics with both biological and statistical motivation. Our contribution is also unique in the significantly larger size of the employed dataset both in terms of number of samples and number of interventions than previous work.

## 3 METHODOLOGY

In the next section, we briefly introduce the formal framework of SCMs in order to serve as a causal language with which to describe methods, limitations and assumptions in this setting in a formal way, as well as to motivate the quantitative metrics presented subsequently. The pertubational nature of the data necessitate a formal statistical language that goes beyond associations and correlations, thus we present and use the causal view as was introduced by Pearl (2009). We will utilise this language to refer back to the setting throughout the rest of the paper.

**Structural Causal Models (SCMs).** In this work, we rely on the language of SCMs to describe causal properties (Peters et al. (2017)). Formally, a SCM $\mathbb{M}$ consists of a 4-tuple $(\mathbb{U}, \mathbb{X}, \mathcal{F}, P(\mathbf{u}))$, where $\mathbb{U}$ is a set of unobserved (latent) variables and $\mathbb{X}$ is the set of observed (measured) variables. $\mathcal{F}$ is a set of function such that for each $X_i \in \mathbb{X}$, $X_i \leftarrow f_i(Pa_i, U_i), U_i \in \mathbb{U}$ and $Pa_i \in \mathbb{X} \setminus X_i$. The SCM induces a distribution over the observed variables $P(\mathbf{x})$, where the uncertainty stems from the distribution on the unobserved variables $P(\mathbf{u})$. The variable-parent relationships can be represented in a directed graph, where each $X_i$ is a node in the graph, and there is a directed edge between all $Pa_i$ to $X_i$. The task of causal discovery can then be described as learning this graph over the variables. The language of causality and in particular the formalism of SCM allows to precisely describe the notion of an intervention in a system. In the most general sense, an intervention on a variable $X_i$ can be thought as uniformly replacing its structural assignment with a new function $X_i \leftarrow \tilde{f}(X_{\widetilde{Pa_i}}, \tilde{U}_i)$. For simplicity, we often consider only simple new functional assignment, such as atomic (constant function) and stochastic (the function only depends on $\tilde{U}_i$). In this work, we only consider such atomic or stochastic intervention and denote an intervention on $X_i$ as $\sigma(X_i)$. We can then describe the interventional distribution, denoted $P^{\sigma(X_i)}(\mathbf{x})$, as the distribution entailed by the modified SCM, where only the assignment for $X_i$ is changed and the other assignment are kept similar to the observational distribution. For consistency, we denote the observational distribution as both $P^{\emptyset}(\mathbf{x})$ and $P(\mathbf{x})$.

For the rest of this paper, we will utilise the SCM framework to describe methods, assumptions and setting.

**Problem Setting.** We consider the setting where we are given a dataset of vector samples $\mathbf{x} \in \mathbf{R}^d$, where $\mathbf{x}_i$ represents the measured expression of gene $i$ in a given cell. Modern gene editing technology (e.g., CRISPR) performs direct interventions on individual genes to bring their expression towards $0$ in case of full on-target efficacy. Given the stochastic nature of cell biology, it is not guaranteed that this results in a $0$ expression, i.e, the interventions are stochastic even though they are close to atomic at level $0$. The goal of a causal model is to learn a causal DAG $\mathcal{G}$, where each node is a single gene. The causal DAG $\mathcal{G}$ induces a distribution over observed sample $P(\mathbf{x})$, such that:

$$P(\mathbf{x}) = \prod p(x_i | Pa_i) \tag{1}$$

The datasets we consider for the benchmark can be thought as consisting of data sampled from $P^{\emptyset}(\mathbf{x})$, as well as $P^{\sigma(X_i)}(\mathbf{x})$ for various $i$. We artificially create different training conditions, namely observational, partially interventional and interventional. The observational setting is restricted to samples $\mathbf{x} \sim P^{\emptyset}(\mathbf{x})$. The interventional settings contains the observational settings, as well as interventional data $\mathbf{x} \sim P^{\sigma(X_i)}(\mathbf{x})$ for all variables $i$. The partial interventional setting is a middle ground between the above two, where only a fraction of the variables have interventional data associated to them, i.e, where they are the target of the intervention.

**Assumptions and challenges.** Single-cell data presents idiosyncratic challenges that may break the common assumptions of many existing methods. Apart from the high-dimensionality in terms of number of variables and large sample size, the distribution of the gene expression present a challenge as it is highly tailed at $0$ for some genes (Tracy et al. (2019)). Regarding the underlying true causal graph, biological feedback loops may break the acyclicity assumption underlying the use of DAGs. In addition, different cells sampled from the same batch may not be truly independent as the cells may have interacted and influenced their states. Lastly, cells in scRNAseq experiments are measured at a fixed point in time and may therefore have been sampled at various points in their developmental trajectory or cell cycle (Kowalczyk et al. (2015)) - making sampling time a potential confounding factor in any analysis of scRNAseq data.

**Network inference methods.** The benchmarked methods are given data, either only observational or also interventional - depending on the setting - consisting of the expression of each gene in each cell. For interventional data, the target gene in each cell is also given as input. We do not enforce that the methods need to learn a graph on all the variables, and further preprocessing and variables selection is permitted. The only expected output is a list of gene pairs that represent directed edges. No properties of the output network, such as for example acyclicity, are enforced

either. As such, even though in this first version of the benchmark, we mainly evaluate causal methods, our benchmark can be used to evaluate any graph outputs without restrictions.

**Evaluating inferred networks.** Contrary to other benchmarks where the true underlying graph is either known, or the data is sampled from a simulated graph, given the nature of inferring causal graphs from real-world experimental data, the true causal graph is unknown in CausalBench as the data is collected from a complex biological process. The lack of a gold standard ground truth network brings about unique challenges in establishing a weight of evidence that supports the use of one method over another: In this setting, there is no certitude regarding the existence of a universal causal graph, that would hold across time points, conditions as well as cell types. To address this challenge, we aim at developing a collection of synergistic metrics for which a higher performance across their set can be assumed to correlate with how close the outputted network is to a true graph. To this end, we implement two types of evaluation: a biologically-motivated and a quantitative statistical evaluation of inferred networks.

**Biologically-motivated evaluation.** The biologically-motivated evaluation in CausalBench is based on biological databases of known putatively causal gene-gene interactions. Using these databases of domain knowledge, we can construct putatively true undirected subnetworks to evaluate the output networks in the understanding that the discovered edges that are not present in those databases are not necessarily false positives, and undirected edges may not necessarily be direct edge connections in a causal graph, i.e, a functional interaction between two genes may be mediated by one or more other genes. The underlying assumption behind the biologically-motivated evaluation approach is that methods that are better will, on average across experimental settings, recall more known gene-gene interactions than those that are worse.

**Quantitative statistical evaluation.** In contrast to the biologically-motivated evaluation, the quantitative statistical evaluation in CausalBench is fully data-driven, cell-specific and prior free. It's also a novel form of network evaluation that is unique to single-cell perturbational data and presents a new approach for approximating ground truth to augment the information contatined in biological databases. To conduct the statistical evaluation, we rely on the interventional data at our disposal in perturbational scRNAseq data to evaluate edges in the output networks. The main assumption for this evaluation is that if the discovered edge from $A$ to $B$ is a true edge denoting a functional interaction between the two genes, then perturbing $A$ should have a statistically significant effect on the distribution $P^{\sigma(X_A)}(\mathbf{x}_B)$ of values that gene B takes in the transcription profile given enough observed samples, compared to its observational distribution $P^{\emptyset}(\mathbf{x}_B)$ (i.e, compared to control samples where no gene was perturbed). The significance of this change in distribution can be tested with statistical test as well as with distributional distances. One of the challenges in using a statistical approach is that the interventional data may be limited in size in practice, which means that some signal may not be significantly detectable with finite data. Lastly, the quantitative statistical approach cannot differentiate between causal effects from direct edges or from causal paths in the graph. Despite those limitations, this quantitative evaluation complements the biologically-motivated evaluation, by extending the covered metric set with metrics that are directly derived from the observed data, making this evaluation data-driven, cell-specific and prior free. Moreover, it presents a novel approach for estimating ground-truth gene regulatory interactions that is uniquely made possible through the size and interventional nature of single-cell perturbational datasets. Full descriptions of both the biologically-motivated and quantitative statistical evaluations are presented in Section 4.

## 4 EXPERIMENTAL EVALUATION

**Datasets.** For the experimental evaluation in CausalBench, we rely on the recently published large-scale scRNAseq dataset by Replogle et al. (2022). Replogle et al. (2022) performed a genome-scale perturb-seq screen, targeting all expressed genes across millions of human cells with CRISPR perturbations. They generated three datasets, one genome-scale pertubational dataset consisting of 2.5 million K562 cells, and two smaller ones, on the K562 (Andersson et al. (1979)) and the RPE1 cell lines, targeting only DepMap essential genes (Tsherniak et al. (2017)). We utilize the last two smaller datasets, as they focus on putatively important genes, and they allow to fairly compare performance on two distinct cell types. To stabilize evaluation and training, we additionally filter out genes for which there are less than 100 perturbed cells, and we hold out 20% of the data for

| dataset | total samples | # observational samples | # gene interventions |
|---|---|---|---|
| (Replogle et al., 2022) K562 | 310 385 | 10 691 | 1 158 |
| (Replogle et al., 2022) RPE1 | 247 914 | 11 485 | 651 |

Table 1: High-level description of the two large-scale datasets utilised in CausalBench - characterised by their high numbers of samples and of intervened-upon variables. Of those samples, after stratification by intervention target (including no target), $20\%$ were kept has held-out data.

evaluation, stratified by intervention target. A summary of the resulting two datasets can be found in Table 1.

**Metrics.** To implement the biologically-motivated evaluation, we extract network data from two widely used open biological databases: CORUM (Giurgiu et al., 2019) and STRING (Von Mering et al., 2005; Snel et al., 2000; Von Mering et al., 2007; Jensen et al., 2009; Mering et al., 2003; Szklarczyk et al., 2010; 2015; 2016; 2019; 2021; Franceschini et al., 2012; 2016). CORUM is a repository of experimentally characterized protein complexes from mammalian organisms. The complexes are extracted from individual experimental publications, and exclude results from high-throughput experiments. We extract the human protein complexes from the CORUM repository and aggregate them to form a network of genes. STRING is a repository of known and predicted protein-protein interactions. STRING contains both physical (direct) interactions and indirect (functional) interactions that we use to create two evaluation networks from STRING: protein-protein interactions (network) and protein-protein interactions (physical). protein-protein interactions (physical) contains only physical interactions, whereas protein-protein interactions (network) contains all types of know and predicted interactions. STRING, and in particular string-network, can contain less reliable links, as the content of the database is pulled from a variety of evidence, such as high-throughput lab experiments, (conserved) co-expression, and text-mining of the literature.

To implement the quantitative evaluation, we leverage the hold out observational and interventional data and compute one statistical test and one distribution distance. For each edge $A$ to $B$, we extract the observational samples for $B$, and the interventional samples for $B$ where $A$ was targeted. We then perform a two-sided Mann-Whitney U rank test (Mann & Whitney, 1947; Bucchianico, 1999) using the SciPy package (Virtanen et al., 2020) to test the null hyptothesis that the two distributions are equal. We count an edge as true positive if the null hypothesis of the statistical test is rejected. To improve statistical power, given that the distribution are heavy tailed towards $0.0$ expression, we trim $90\%$ of the $0.0$ values of the observational distribution, and the same fraction of the smallest values of the interventional samples. For the distributional distance, we compute a Wasserstein distance (Ramdas et al., 2017) between the two empirical distributions. We then return the mean Wasserstein distance of all the inferred edges. We do not normalize the computed distance, but we expect this to not have a significant effect as the samples always have more or less the same size (exactly the same for the observational samples, and similar for interventional, and at least 20 by design), The advantage compared to a statistical test is that we obtain a continuous metric that correlate with the strength of causal effect of parent nodes on children nodes and does not need choosing a hyperparameter such as the $p$-value threshold. A higher mean Wasserstein distance would then indicate better performance.

**Baselines.** We implement a representative set of existing state-of-the-art methods for the task of causal discovery from single-cell observation and mixed perturbational data. For the observational setting, we implement PC (named after the inventors, Peter and Clark; a constraint-based method) (Spirtes et al., 2000), Greedy Equivalence Search (GES; a score-based method) (Chickering, 2002), and NOTEARS variants NOTEARS (Linear), NOTEARS (Linear,L1), NOTEARS (MLP) and NOTEARS (MLP,L1) (Zheng et al., 2018; 2020). In the interventional setting, we included Greedy Interventional Equivalence Search (GIES, a score-based method and extension to GES) (Hauser & Bühlmann, 2012), and the Differentiable Causal Discovery from Interventional Data (DCDI) variants DCDI-G and DCDI-DSF (continuous optimization based) (Brouillard et al., 2020). GES and GIES greedily add and remove edges until a score on the graph is maximized. NOTEARS and DCDI enforce acyclicity via a continuously differentiable constraint, making them suitable for deep learning. We provide a broad overview of method classes included below:

*PC* is one of the most widely used methods in causal inference from observational data that assumes there are no confounders and calculates conditional independence to give asymptotically correct

results. It outputs the equivalence class of graphs that conform with the results of the conditional independence tests.

*Greedy Equivalence Search (GES)* implements a two-phase procedure (Forward and Backward phases that adds and removes edges from the graph) to calculate a score to choose within an equivalence class. While GES leverages only observational data, its extension, Greedy Interventional Equivalence Search (GIES), enhances GES by adding a turning phase to allow for the inclusion of interventional data.

*NOTEARS* formulates the DAG inference problem as a continuous optimization over real-valued matrices that avoids the combinatorial search over acyclic graphs. This is achieved by constructing a smooth function with computable derivatives over the adjacent matrices that vanishes only when the associated graph is acyclic. Various versions of NOTEARS refer to which function approximator is employed (either a MLP or Linear) or which regularity term is added to the loss function (e.g., L1 for the sparsity constraint.)

*Differentiable Causal Discovery from Interventional Data (DCDI)* (Brouillard et al., 2020) leverages various types of interventions (perfect, imperfect, unknown), and uses a neural network model to capture conditional densities. DCDI encodes the DAG using a binary adjacency matrix. The intervention matrix is also modeled as a binary mask that determines which nodes are the target of intervention. A likelihood-based differentiable objective function is formed by using this parameterization, and subsequently maximized by gradient-based methods to infer the underlying DAG. DCDI-G assumes Gaussian conditional distributions while DCDI-DSF lifts this assumption by using normalizing flows to capture flexible distributions.

*GRNBoost* (Aibar et al., 2017) is a GRN specific Gradient Boosting tree method, where for every gene, candidate parent gene are ranked based on their predictive power toward the expression profile of the downstream gene. As such, it acts as a feature selection method toward learning the graph. GRNBoost was identified as one of the best performing GRN method in previous observational data based benchmarks (Pratapa et al., 2020).

*Random (k)* is the simplest baseline which outputs a graph from which $k$ nodes are selected at uniformly random without replacement. In the experiments, we tested $k = 100, 1000$, and $10000$.

Unfortunately, all of the above-mentioned methods, with the exception of NOTEARS and GRNBoost, do not computationally scale to graphs of the size typically encountered in transcriptome-wide gene regulatory network inference. To nonetheless enable a meaningful comparison, we propose to partition the variables into smaller subsets and run the above-mentioned methods on each subset independently with the final output network being the union of the subnetworks. Acyclicity is guaranteed if each subnetwork is acyclic. The proposed approach breaks the no-latent-confounder assumption that some methods may make, and it also does not fully leverage the available information potentially in the data. Methods that do scale to the full graph in a single optimization loop are therefore expected to perform better.

For all the above state-of-the-art methods, we use publicly available implementations and we implement interfaces between our benchmark suite and the open-source packages. We use causal-learn (Zhang et al., 2022) for PC and GES, the official implementation of the authors for DCDI (Brouillard et al., 2022), a public python implementation for GIES (Gamella, 2022), and the official implementation of the authors for NOTEARS (Zheng et al., 2018; 2020).

## 5 RESULTS AND DISCUSSIONS

**Network inference.** We here summarize the experimental results of our baselines in both observational and interventional settings, on the quantitative evaluation (statistical test) and the biologically-motivated evaluations (corum, string-network and string-physical). All results are obtained by training on the full dataset three times with different random seeds. We highlight the trade-off between true positives count and ~~true positive rate~~ precision. Indeed, we expect methods to optimize for these two goals, as we want to obtain a high ~~true positive rate~~ precision while maximizing the number of discovered interactions. In the observational setting, we observed this trade-off, as some methods can achieve very high ~~true positive rates~~ precision, but with very small output graphs (Figure 2). Also, methods that regularize for sparsity, such as NOTEARS (Linear,L1) and (MLP,L1), have

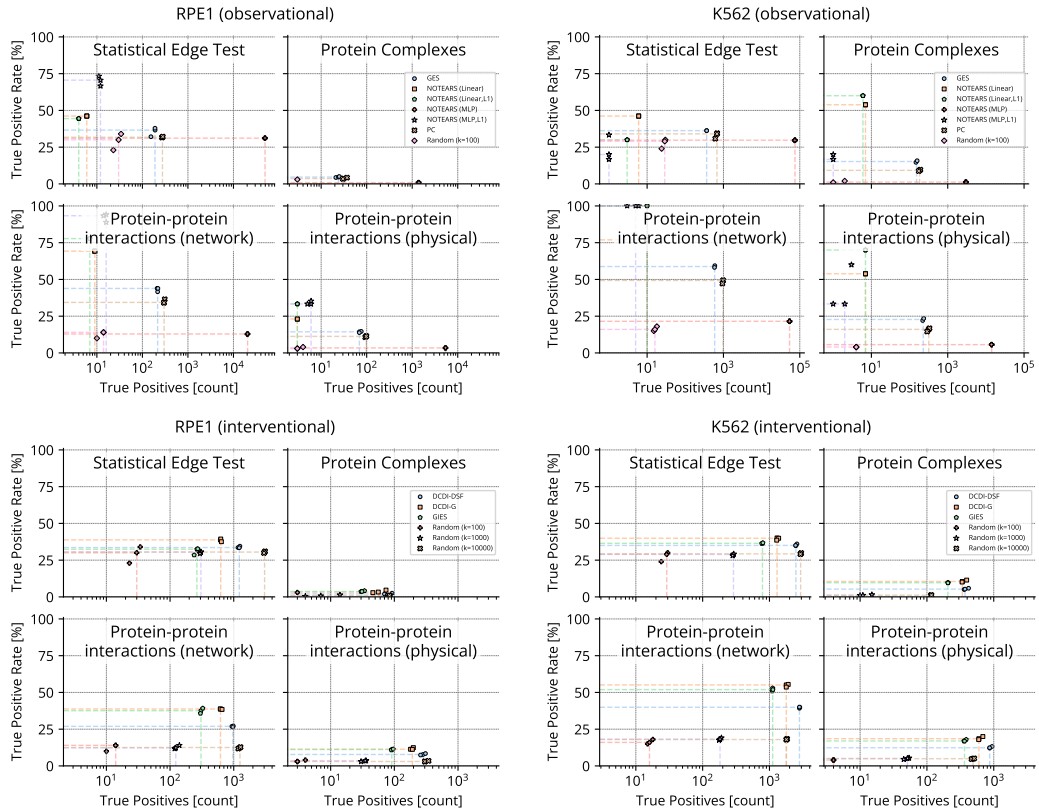

Figure 2: Performance comparison in terms of ~~True Positive Rate~~ (TPR, in %; y-axis) Precision and True Positives (TP, in counts; x-axis) in correctly identifying edges substantiated by four different biological interaction databases, Statistical Edge Test (ours; top left), protein complexes (CORUM, Giurgiu et al. (2019); top right), protein-protein interactions from networked sources (STRING Szklarczyk et al. (2021); bottom left), and protein-protein interactions from physical assays (STRING Szklarczyk et al. (2021); bottom right), between 7 different methods (see legend in top row) using observational (top row) and 6 different methods (see legend in bottom row) using interventional data data (bottom row) in RPE1 (left column) and K562 (right column) cell lines. For each method, we report the results for three independently calculated runs with new random seeds to assess uncertainty. Dashed lines in the same color as the markers indicate the median performance observed across all randomly seeded runs.

higher ~~true positive rates~~ precision in general, which suggests the importance of sparsity when dealing with noisy data. Regarding the quantitative statistical test, ~~satisfactory results are only obtained on the RPE1 dataset~~ high precision (around 75%) is only achieved on the RPE1 dataset as compared to around 50% on the K562 dataset. This could be explained in part by the interventions having greater effects in this cell line, as reported also by Replogle et al. (2022). In the interventional setting, results are similar to the observational setting, but with most methods outputting larger graphs with smaller ~~true positive rates~~ precision. Surprisingly, the difference to the observational setting is not large, especially in the statistical test where access to some of the interventional data should theoretically help. This highlights an opportunity for further method development for causal graph inference in order to be able to fully leverage more interventional data in the future. To offer a more principled way to compare the models across all these metrics, we propose a simple and unbiased way to compute a ranked scoreboard in Appendix C

**Computational performance.** All methods were given the same computational resources, which consists of 30 CPU's with 64GB of memory per each. We additionally assign a GPU for the DCDI methods. The hyperparameters of each method, such as partition sizes, are chosen such that the running time remains below 30 hours. Partition sizes for each model can be found in Appendix B.

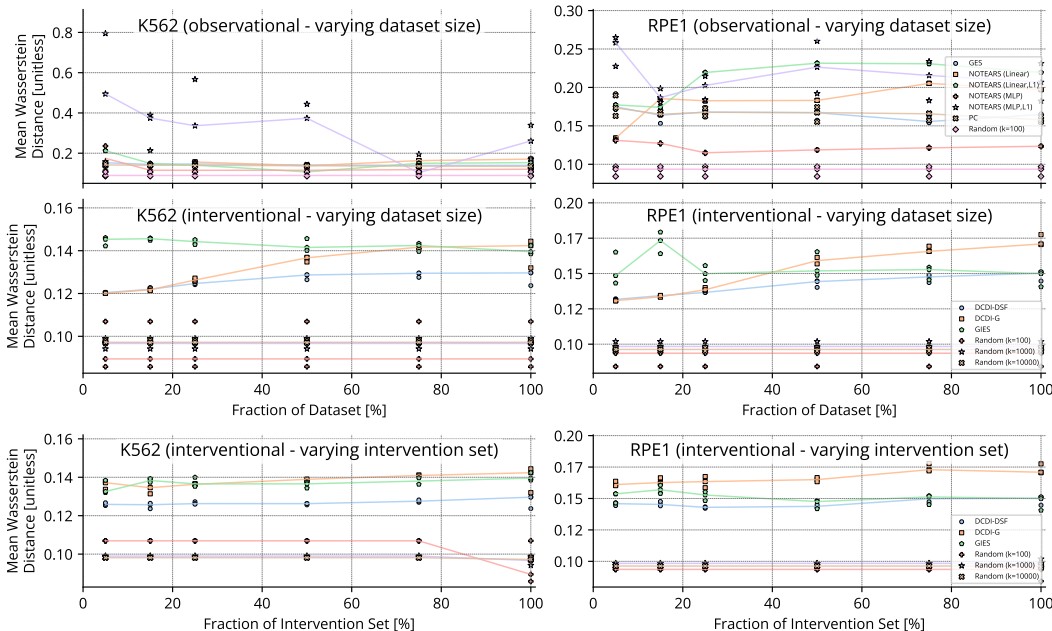

Figure 3: Performance comparison in terms of Mean Wasserstein Distance (unitless; y-axis) of 7 methods for causal graph inference on observational data (top row; (see legend top right) and 6 methods on interventional scRNAseq data (centre row; see legend centre right) when varying the fraction of the full dataset size available for inference (in %; x-axis), and 6 methods on interventional data (bottom row; see legend bottom right) when varying the fraction of the full intervention set used (in %, x-axis). Markers indicate the values observed when running the respective algorithms with one of three random seeds, and colored lines indicate the median value observed across all tested random seeds for a method.

**Performance as a function of sample size.** We additionally studied the effect of sample size on the performance of the evaluated state-of-the-art models. We randomly subset the data at different fractions and report the mean Wasserstein distance as explained in the quantitative evaluations part of Section 3 and shown in Figure 3. In the observational setting, the sample size does not seem to have a significant effect on performance for most methods, indeed having a slightly negative effect for some methods. This suggest that the ability to effectively leverage a large sample size of perturbational data is also an open challenge that needs to be addressed for high performance. In the interventional setting, a positive impact for larger sample size is observable, especially for the methods that rely on deep networks and gradient-based learning such as DCDI, whereas GIES seems to suffer in a large sample setting.

**Performance by the fraction of perturbations.** Beyond the size of the training set, we also studied the partial interventional setting - where only a subset of the possible genes to perturb are experimentally targeted. We adapt the fraction of randomly targeted genes from 5% (low ratio of interventions) to 100% (fully interventional). We randomly subset the genes at different fractions, using three different random seeds for each method, and report the mean Wasserstein distance as a measure of quantitative evaluation. We would expect a larger fraction of intervened genes to lead to higher performance, as this should facilitate the identification of the true causal graph. We indeed observe a better performance for DCDI as we increase the fraction of intervened genes. For GIES, we again observe a negative effect of having more data, probably mainly due to the fact that higher fraction of intervened genes implies a larger number of training samples in total.

**Limitations.** Openly available benchmarks for causal models for large-scale single cell data could potentially accelerate the development of new and effective approaches for uncovering gene regulatory relationships. However, some limitations to this approach remain: firstly, the biological networks used for evaluation do not fully capture ground truth gene regulatory networks, and the reported connection are often biased towards more well studied systems and pathways (Gillis et al. (2014)). True ground truth validation would require prospective and exhaustive interventional wet-lab experiments. However, at present, experiments at the scale necessary to exhaustively map gene-gene interactions across the genome are cost prohibitive for all possible edges even for the largest research institutes in the world and even when not considering cell heterogeneity and the confound-

ing factor of time. We expect more publicly available experimental validation data to aggregate over time, and are looking to incorporate these data into the benchmark in the future. Beyond limitations in data sources used, there are limitations with some of the assumptions in the utilised state-of-the-art models: For instance, feedback loops between genes are a well-known phenomenon in gene regulation (Carthew (2006); Levine & Davidson (2005)) that unfortunately presently cannot be represented by existing causal network inference methods.

## 6 CONCLUSION

We introduced CausalBench - a comprehensive open benchmark for evaluating algorithms for discovering gene regulatory networks using a large-scale CRISPR-based interventional scRNAseq dataset. CausalBench introduces a set of biologically meaningful performance metrics to compare the graphs proposed by causal inference methods quantitatively using statistical tests, and according to existing biological knowledge bases on protein-protein interactions and protein complexes. In addition, CausalBench contains implementations of a range of state-of-the-art causal discovery algorithms from those that rely on independence tests (e.g., the PC algorithm), score-based methods (e.g., GIES) and the more recently introduced continuous optimization-based methods (e.g., different versions of NOTEARS). CausalBench was designed to lower the barrier of entry for developing a causal discovery algorithm for scRNAseq data by fixing non-model-related components of the evaluation pipeline and allow users to focus on the advancement of causal network discovery methods. With benchmark sample sizes of more than 200 000 interventional samples, CausalBench is built on one of the largest open real-world interventional datasets (Replogle et al. (2022)) - ushering in an era in which the community of causal machine learning researchers has ready access to real-world large-scale interventional datasets for developing and evaluating causal discovery methods.

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

## A  BENCHMARK PACKAGE

### A.1  INSTALL REQUIREMENTS

The necessary python package requirements to run the benchmark can be found in the `requirements.txt` file. They can be installed by running the following command:

```
pip install -r requirements.txt
```

### A.2  RUNNING THE BENCHMARK

Here is an example of a command to run a model on the k562 dataset in the observational regime:

```
python3 causalscbench/apps/main_app.py \
--dataset_name weissmann_k562 \
--output_directory /path/to/output/ \
--data_directory /path/to/data/storage \
--training_regime observational \
--model_name model \
--subset_data 1.0 \
--model_seed 0
```

Results are written to the folder at /path/to/output/, and processed datasets will be cached at /path/to/data/storage. See the MainApp class for more hyperparameter options, especially in the (partial) interventional setting.

### A.3  ADDING A MODEL

New models can be easily added to the benchmark suite. The only contract for a model is to implement the [AbstractInferenceModel] class. Here is a simple example where the model returns a fully connected gene-gene graph:

```python
from causalscbench.models.abstract_model import AbstractInferenceModel

class FullyConnected(AbstractInferenceModel):
    def __init__(self) -> None:
        super().__init__()

    def __call__(
```

| Model name | partition size |
|---|---|
| PC | 30 |
| GES | 30 |
| GIES | 30 |
| NOTEARS (Linear) | -1 |
| NOTEARS (Linear, L1) | -1 |
| NOTEARS (MLP) | -1 |
| NOTEARS (MLP, L1) | -1 |
| DCDI-DSF | 50 |
| DCDI-G | 50 |

Table 2: Partition sizes used for each model. $-1$ means that the graph was not partitioned.

```
        self ,
        expression_matrix : np.array ,
        interventions : List [ str ] ,
        gene_names : List [ str ] ,
        training_regime : TrainingRegime ,
        seed : int = 0 ,
    ) -> List [ Tuple ]:
        random.seed ( seed )
        edges = set ()
        for i in range ( len ( gene_names )):
            a = gene_names [ i ]
            for j in range ( i + 1 , len ( gene_names )):
                b = gene_names [ j ]
                edges.add (( a , b ))
                edges.add (( b , a ))
        return list ( edges )
```

The new model then only needs to be included in the `MainApp` model registry.

## B    PARTITION SIZES

We here recapitulate the partition sizes used to be able to run each methods in Table 2.

## C    MODEL RANKING

We here present a simple unbiased way of ranking the different models based on all the implemented evaluation metrics. We separate the rankings per cell type. First, we create a preliminary ranking for each evaluation metrics. For metrics based on precision-recall, we compute a score by taking the mean of precision and recall (for Statistical, Protein Complexes, PPI network, PPI physical and CHIP-seq network evaluations). The weighting may be modulated, for example depending on the downstream application. Finally, for each model, we take their average rank across the evaluation specific rankings. This ranking thus gives the same weight to each evaluation method. Results are summarized in Table 3 for the K562 cell line and in Table 4 for the RPE1 cell line. As can be observed, methods able to scale to the full graph (GRNBoost and NOTEARS) perform the best.

## D    RUN TIME

We here present the average run time for each method in Table 5 for the K562 cell line and in Table 6 for the RPE1 cell line

| Model | Average rank |
|---|---|
| Random (k = 100) | 10.83 |
| Random (k = 1000) | 10.5 |
| Random (k = 10000) | 8.66 |
| PC | 6.66 |
| DCDI-DSF | 6.5 |
| NOTEARS (MLP, L1) | 6.0 |
| DCDI-G | 5.5 |
| GES | 5.16 |
| GIES | 4.83 |
| NOTEARS (Linear) | 4.16 |
| NOTEARS (Linear, L1) | 3.66 |
| NOTEARS (MLP) | 3.33 |
| GRNBoost | 2.16 |

Table 3: Ranking of the methods on the K562 cell line. Methods able to scale to the full graph (GRNBoost and NOTEARS) perform the best.

| Model | Average rank |
|---|---|
| Random (k = 100) | 10.5 |
| Random (k = 1000) | 9.99 |
| Random (k = 10000) | 8.66 |
| GIES | 8.33 |
| PC | 7.66 |
| GES | 6.33 |
| NOTEARS (MLP, L1) | 5.16 |
| NOTEARS (Linear) | 5.0 |
| DCDI-DSF | 4.16 |
| NOTEARS (Linear, L1) | 3.83 |
| DCDI-G | 3.66 |
| NOTEARS (MLP) | 2.83 |
| GRNBoost | 1.83 |

Table 4: Ranking of the methods on the RPE1 cell line. Methods able to scale to the full graph (GRNBoost and NOTEARS), as well as the DCDI methods, perform the best.

| Model | Run time (hours) |
|---|---|
| DCDI-DSF | 60.15 |
| DCDI-G | 30.49 |
| NOTEARS (MLP) | 27.78 |
| NOTEARS (Linear) | 14.89 |
| PC | 9.58 |
| GES | 5.79 |
| GIES | 3.97 |
| NOTEARS (Linear, L1) | 2.15 |
| NOTEARS (MLP, L1) | 0.82 |
| GRNBoost | 0.3 |
| Random (k = 10000) | 0.0 |
| Random (k = 1000) | 0.0 |
| Random (k = 100) | 0.0 |

Table 5: Run time in hours for each method on the K562 cell line. GRNBoost was run on the observational data only.

| Model | Run time (hours) |
|---|---|
| DCDI-DSF | 37.52 |
| DCDI-G | 19.68 |
| PC | 13.18 |
| NOTEARS (MLP) | 10.9 |
| NOTEARS (Linear) | 3.7 |
| GES | 3.22 |
| NOTEARS (Linear, L1) | 0.96 |
| NOTEARS (MLP, L1) | 0.62 |
| GIES | 0.58 |
| GRNBoost | 0.09 |
| Random (k = 10000) | 0.0 |
| Random (k = 1000) | 0.0 |
| Random (k = 100) | 0.0 |

Table 6: Run time in hours for each method on the RPE1 cell line. GRNBoost was run on the observational data only.

