# OpenReview forum: "CausalBench: A Large-scale Benchmark for Network Inference from Single-cell Perturbation Data"
_ICLR.cc/2023/Conference — Submitted to ICLR 2023_

### Official Review · Reviewer_2A85 · 2022-10-21

**Confidence:** 4
**Correctness:** 3
**Technical Novelty And Significance:** 2
**Empirical Novelty And Significance:** 3
**Recommendation:** 5

**Clarity, Quality, Novelty And Reproducibility:**

The paper is clear and easy to read and understand. The novelty is only limited to introducing a a benchmarking bed for causal gene network inference methods using perturbational scRNAseq data.

**Strength And Weaknesses:**

Strength:

- This paper tries to take the first step towards benchmarking the causal gene regulatory network inference methods using large-scale perturbational scRNA-seq data, which is interesting and important for causal discovery in biology.

Weakness:

- The idea of having a framework to test the causal network inference methods is great if the benchmarking metrics are powerful and interesting enough. I think the metrics proposed in this paper are not that much interesting. For the biological metrics, as mentioned by the authors, the well-known gene-gene or protein-protein interaction data are not specific for the cell types that they are working. The underlying gene regulation rules are very cell-type specific and the inferred networks could not be reliably evaluated by general networks which have been derived from many cell types.

**Summary Of The Paper:**

This paper introduces CausalBench, a framework to benchmark gene regulatory inference methods using perturbational scRNAseq data. Although having access to perturbational scRNA data can help in discovering causal gene networks, generating large-scale perturbational data where every gene undergoes some genetic intervention (knockdown) is very cost prohibitive. There is a recent study that generates genome-wide perturbational scRNA data in only two cell types. The authors of CausalBench have used these data to benchmark some of the well-known network inference methods which use observational and interventional data. For the evaluation, they propose two approaches: biological and quantitative. For the biological evaluation, they use known gene-gene or protein-protein interaction data, and for the quantitative, they use some statistical tests to validate the effects of interventions.

**Summary Of The Review:**

Causal gene network inference (DAGs) are very important in understanding the mechanisms of gene expression. However, learning such DAGs using only observational data is not possible. Recent Perturb-seq data provide helpful interventional data by genetically perturbing the target genes and measuring the expression of other genes. This data empowers the causal DAG learning methods, however, they should be tested and benchmarked via the same metrics to measure their efficacy.

This paper takes the first step in the right direction by noticing the need for benchmarking causal network inference methods using Perturb-seq data. However, the benchmarking metrics are not comprehensive enough to be certain about the efficacy of such methods. My comments:

- Biological metric does not make so much sense to me. The underlying DAG driving the gene expression programs are very cell-type specific. So, the general gene/protein networks cannot be a good and reasonable metric to evaluate the correctness of the learned DAGs in K562 and RPE1 cells.

- As a qualitative metric, the authors can use held out perturbations and see how well different methods can predict the effect of each perturbation on various genes. This has also been used in Lopez et al. (2022) (cited in the paper).

- Why haven't the authors used the biggest Perturb-seq (2 M cells) in K562 cells as published in (Replogle et al., 2022)? It would be interesting to see how well/fast each method works in a very large dataset.

- Related to the previous comment, I think one another and yet interesting benchmarking metric would be how fast each method could be trained in each dataset. This kind of benchmarking has also been done in Lopez et al. (2022). By growing sequencing technology, it is not so far that we will have access to millions of cells and thousands of perturbations genome-wide, and the methods that could handle this huge computational barrier would be the winners.

Minor commets:

Typos:   - we we observed --> we observed
             - in not large --> is not large

---

> ### Author Response · Authors · 2022-11-19
> **Authors Response (Part 1) to Official Review of Paper5185 by Reviewer 2A85**
>
> Thanks for your feedback and for recognizing our work as a first step towards benchmarking GRN inference using large-scale perturbational scRNA-seq data and pointing out its importance. We address each of your concerns below. In case an answer requires a change in the manuscript, the change is colored red in the revised manuscript for better visibility
>
> C1: Biological metric does not make so much sense to me. The underlying DAG driving the gene expression programs are very cell-type specific. So, the general gene/protein networks cannot be a good and reasonable metric to evaluate the correctness of the learned DAGs in K562 and RPE1 cells.
>
> AR: We agree that it is important to account for cell type-specific effects when evaluating gene regulatory network inference methods. Following the recommendation made by reviewer vpJf, we have adjusted our biological evaluation to incorporate cell-type specific networks. In concordance with existing literature on the subject, we make use of the STRING protein-protein interaction network as well as a ChiPSeq-derived network of transcription factor binding relationships.
>
> In the case of K562, we used Chip-Atlas and ENCODE databases to limit the links in the ChipSeq networks to only those that are relevant for or have been measured in the K562 cell line. However, in the case of RPE-1, such an extensive level of experimental characterization has not been performed in the literature. In this case, we make use of similarly derived networks but for another epithelial cell line (since RPE-1 is an epithelial cell) that is more well-studied (HepG2).
>
> Further, we would also like to direct you to our statistical evaluation which is based on held-out perturbations from the same cell type that is used for training the model, thus making it cell-type specific. We are the first benchmark to leverage perturbation data for the inference of gene regulatory networks. This allows us to measure performance (through the statistical evaluation metric) against a larger space of putative gene regulatory interactions than would be possible using biological databases alone. Moreover, network inference methods trained on interventional data should theoretically come closer to estimating the true underlying causal graph since interventional data is better able to restrict the size of the Markov equivalence classes as compared to existing benchmarks based on observational data alone (Pratap et al. 2020).
>
> C2: As a qualitative metric, the authors can use held out perturbations and see how well different methods can predict the effect of each perturbation on various genes. This has also been used in Lopez et al. (2022) (cited in the paper).
>
> AR: Using hold-out perturbations to validate model predictions for unseen links using perturbational effects is indeed a useful strategy given the size and scale of this dataset. We have applied this approach in our statistical evaluation procedure which involves holding out some perturbation X and predicting links between the targeted gene and other downstream genes. These links are then evaluated using the held out ground-truth perturbational outcome data on whether the perturbation of X resulted in a statistically significant change in the expression of that downstream gene.

---

> > ### Comment · Reviewer_2A85 · 2022-11-23
> > **Response**
> >
> > Thank you for answering my questions. My concerns regarding the the rigor of the biological benchmarks still remain open. Furthermore, the statistical benchmark is not comprehensive in that it only considers the effect (direct or indirect) of perturbing one gene, while we could have multiple genes perturbed at the same time. My score remains the same.

---

> > > ### Author Response · Authors · 2022-11-30
> > > **Authors response**
> > >
> > > Thank you for your prompt response. We very much appreciate the provided feedback.
> > >
> > > Reviewer 2A85: "My concerns regarding the the rigor of the biological benchmarks still remain open."
> > >
> > > AR: Thank you very much for the clarification. To address your concerns around the biological networks not being cell-type specific, we have added the CHIP-Seq reference to the revised manuscript, which is a cell-type specific network (Pratapa et al. 2020) that matches the K562 cell line precisely, and the RPE-1 cell line through another epithelial cell for which CHIP-Seq evidence was available (HepG2).
> > > In addition, we would like to reiterate that the best performance on the benchmark can only be achieved by performing well on both the biological *and* the statistical evaluation, which is specific to the perturbation data generated and hence also specific to the cell-type considered in the experiment. Given the causal nature of the gene-regulatory network inference task, it is unfortunately fundamentally not possible to have perfect ground truth in real-world data - the premise of CausalBench is to therefore compare to the next best thing: to compare the findings to the best characterised biological knowledge (the biological evaluation) and the statistical evidence generated in the experiment itself (the statistical evaluation). This approach matches the intuitive strategy of scoring highly generated networks that are consistent with both known information and the available statistical evidence. Given it is fundamentally not possible to generate perfect ground truth corresponding to the given experimental setting, we would be keen to better understand Reviewer 2A85 ask for additional rigor: Concretely, is your suggestion (i) to not evaluate against real world-data because no ground truth is available or (ii) to not utilise non-cell-type specific information for benchmarking?
> > >
> > > If your suggestion is (ii), we would be more than happy to make available a version of the benchmark that does not utilise non-cell-type specific information. If your suggestion is (i), we would argue that this is not a feasible path forward for accurately benchmarking progress in gene regulatory network inference algorithms - the intricacies of single-cell RNAseq data are well characterised (Chen, G., Ning, B., & Shi, T. (2019); Hicks, S. C., Teng, M., & Irizarry, R. A. (2015)) and purely simulated data (the only viable alternative with ground truth data available) do not accurately reflect the real-world data that algorithms are subjected to in practice (Crowell, H. L., Leonardo, S. X. M., Soneson, C., & Robinson, M. D. (2021)). We would be happy to receive your clarification.
> > >
> > > Reviewer 2A85: "The statistical benchmark is not comprehensive in that it only considers the effect (direct or indirect) of perturbing one gene, while we could have multiple genes perturbed at the same time."
> > >
> > > Reviewer 2A85 is correct that CausalBench is a framework for evaluating network inference algorithms on single-perturbation single-cell RNAseq datasets. We will be happy to more prominently state the addressed single-perturbation setting in the manuscript.
> > > Please note that the single-perturbation setting (1) is the most important one both for practical purposes (drugs are predominantly developed to target a single molecular target), (2) the most important one for better understanding gene function (genes are typically studied biologically on the level of a single gene to avoid confounding with the effect of another gene's function), and (3) the setting for which the most extensive data is available in form of scRNAseq studies.
> > >
> > > We would be happy to receive your clarifications in light of the above. We would also like to reiterate that we remain open to and committed to implementing all changes suggested to address your remaining concerns. Many thanks in advance for your kind support.

---

> ### Author Response · Authors · 2022-11-19
> **Authors Response (Part 2) to Official Review of Paper5185 by Reviewer 2A85**
>
> C3: Why haven't the authors used the biggest Perturb-seq (2 M cells) in K562 cells as published in (Replogle et al., 2022)? It would be interesting to see how well/fast each method works in a very large dataset.
>
> AR: We agree with you that testing network inference on the full Replogle Perturb-seq dataset (2 million cells) is a very exciting direction. Causalbench is a forward-looking benchmark, since we are providing a framework for evaluating network inference procedures using large-scale gene perturbation experiments which are likely to become more and more ubiquitous over time. CRISPR-based interventional screens present the most economically viable strategy for inferring regulatory relationships between genes at a large scale.
>
> At present, based on our benchmarking results, existing causal inference approaches struggle with scaling to even the smaller subset of perturbations (1158) that we consider in our paper. Similar results have also been reported by other papers that have attempted to use this dataset or other datasets as large as this one [1, 2]. Another reason for making this choice was because this resulted in roughly the same number of perturbations for both K562 and RPE1 and enabled an easier comparison across cell lines which can be helpful for further validation.
>
> However, we are mindful of the rapid developments ongoing in the field and will update our benchmark to allow for the evaluation for the full 2M cells dataset. We are unable to provide a baseline for the full larger dataset but can evaluate predicted GRNs.
>
> [1] Lopez, R., Hütter, J. C., Pritchard, J. K., & Regev, A. (2022). Large-scale differentiable causal discovery of factor graphs. arXiv preprint arXiv:2206.07824.
>
> [2] Lopez, R., Tagasovska, N., Ra, S., Cho, K., Pritchard, J. K., & Regev, A. (2022). Learning Causal Representations of Single Cells via Sparse Mechanism Shift Modeling. arXiv preprint arXiv:2211.03553.
>
>
> C4: Related to the previous comment, I think one another and yet interesting benchmarking metric would be how fast each method could be trained in each dataset. This kind of benchmarking has also been done in Lopez et al. (2022). By growing sequencing technology, it is not so far that we will have access to millions of cells and thousands of perturbations genome-wide, and the methods that could handle this huge computational barrier would be the winners.
>
> AR: The benchmark has been updated to record run time. We nevertheless do not use this as a metric as we consider output quality to be more important than run time, which can also be hardware dependent. Please see the Appendix D for details.
>
> C5: we we observed --> we observed - in not large --> is not large
>
> AR: Thanks! We have corrected this and other typos that we noticed in the text post-submission.

---

### Official Review · Reviewer_vpJf · 2022-10-24

**Confidence:** 4
**Correctness:** 3
**Technical Novelty And Significance:** 2
**Empirical Novelty And Significance:** 3
**Recommendation:** 6

**Clarity, Quality, Novelty And Reproducibility:**

The presented work is of good quality and is overall clearly explained, allowing reproducibility of the results. Regarding originality, some of the metrics that the authors are proposing are novel but others have been previously used. Moreover, while it is good to propose novel metrics, the authors miss to include some of the classical ones, which would make comparisons to previous work easier.


**Strength And Weaknesses:**

Strengths:
- Authors are aware of and nicely describe the assumptions and limitations of this evaluation task and data.
- Including both biological and statistical metrics for evaluation of method performance.
- Evaluation frameworks allow to easily test if novel methods improve over baselines, motivating the development of improved methods.

Weaknesses:
- Authors propose novel evaluation metrics but did not implement classical ones such as the comparison against ChIP-seq derived networks, see (Pratapa et al. 2020).
- Authors miss to evaluate classic methods, such as SCENIC (Aibar et al. 2017), that have been shown to be the top performers in other benchmarks (Pratapa et al. 2020). Although these methods are not causal, it would be beneficial to assess if the modeling of causality actually improves GRN inference.
- While the authors provide different metrics, it is still not clear which methods perform better than others consistently. A “consensus” score based on rankings would improve interpretation. Moreover, there is no discussion about which methods overperform the others and why.
- Although the authors are aware of the limitation of perturbation experiments, no quality control assessment is performed to test whether the perturbation actually worked. Samples where the perturbed gene still shows high levels of gene expression should be removed or at least accounted for.
- The partition of the data due to scalability issues raises concerns since, as the authors already mention, it breaks the no-latent-confounder assumption. Authors chose partition sizes such that the running time remains below 30 hours but do not mention anywhere the actual numbers for each method. In order to comply with this rule, some methods might use fractions of the feature space that are too small to generate valuable graphs.
- In figure 3, authors mention “significant” increase but do not perform any statistical test to corroborate it.



**Summary Of The Paper:**

The authors present CausalBench, a framework to benchmark causal gene regulatory network (GRN) inference methods on perturbational single-cell RNA sequencing data (scRNA-seq). It includes evaluation metrics, baseline implementations of relevant inference methods and access to perturbational scRNA-seq data. With CausalBench the authors evaluate the ability of recovering “silver-standard” ground truth networks and how they are affected by sample size.

**Summary Of The Review:**

The author's contributions to the GRN evaluation field are somewhat new but some aspects already exist in previous work. Moreover, the authors are missing some previous contributions, limiting the added cumulative value of it. On the other hand, the description of the problem, its assumptions and the description of the methods is overall clear and of good quality.

---

> ### Author Response · Authors · 2022-11-19
> **Authors Response (Part 1) to Official Review of Paper5185 by Reviewer vpJf**
>
> Thank you for acknowledging various aspects of this work such as including both biological (and truly pointed out some of which are novel) and statistical evaluation metrics in the benchmark and also pointing out the potential applications of the benchmark framework for scoring the network inference algorithms. We address each of your concerns below. In case an answer requires a change in the manuscript, the change is colored red in the revised manuscript for better visibility
> (Cx: Reviewer’s Concern number x, AR: Author Response)
>
>
> C1: Authors propose novel evaluation metrics but did not implement classical ones such as the comparison against ChIP-seq derived networks, see (Pratapa et al. 2020).
>
> AR: We thank the reviewer for the suggestion. We have extended the biological evaluation accordingly with two cell-specific directed ChIP-seq-derived networks.
>
>
> C2: Authors miss to evaluate classic methods, such as SCENIC (Aibar et al. 2017), that have been shown to be the top performers in other benchmarks (Pratapa et al. 2020). Although these methods are not causal, it would be beneficial to assess if the modeling of causality actually improves GRN inference.
>
> AR: Even though our benchmark is mainly targeted at causal methods, it does not impose any restriction on the type of methods used for GRN inference except for being data-driven and prior-free.Thus, acknowledging your suggestion, we have GRNBoost as a baseline model, which is the backend of SCENIC (when not including gene sequence information), and will report its performance. We chose GRNBoost since it was found to be the best-performing classical approach in the BEELINE benchmark [1]. As can be observed, GRNBoost outperforms most causal-based methods, highlighting the need for additional work needed to translate causal models toward solving the task of gene interaction network inference.
>
> We also want to highlight another factor that distinguishes CausalBench from classical methods or more recent methods, which is being able to leverage gene perturbational data for GRN inference. This allows us to come closer to estimating the underlying causal graph since network inference methods trained on interventional data should theoretically better restrict the size of the Markov equivalence classes as compared to existing benchmarks based on observational data alone (e.g. [1]). We believe that CausalBench is a forward-looking gene regulatory network inference benchmark since large-scale gene perturbation experiments are likely to become more and more ubiquitous over time and present the most economically viable strategy for inferring regulatory relationships at scale.
>
> [1] Pratapa, Aditya, et al. "Benchmarking algorithms for gene regulatory network inference from single-cell transcriptomic data." Nature methods 17.2 (2020): 147-154.
>
> C3: While the authors provide different metrics, it is still not clear which methods perform better than others consistently. A “consensus” score based on rankings would improve interpretation. Moreover, there is no discussion about which methods overperform the others and why.
>
> AR: Thank you for raising these concerns. Our goal in this paper was to build a software system with embedded evaluation, data loaders, and other necessary tools so that a user can try out a method easily and compare its performance with the existing ones according to a number of included metrics which can also be extended effortlessly. We did not attempt to identify one method as the best-performing one since we felt that defining an aggregate metric might be misleading as methods that work well on average may still be sub-optimal for a particular experimental setting. However, we understand the reviewer’s concerns and present a potential aggregate metric in the following.
> We use two sources of evidence for validation:
> 1) The biological evidence that comes from the accumulated knowledge of biology in the literature.
> 2) The statistical test that makes plausible but minimal assumptions about the underlying GRNs and uses only the available data to validate it.
>
> As these sources are of a completely different nature, weighing them in an aggregate metric requires an assumption about their relative importance which consequently inject more bias into the evaluation method. As a proof of concept, we propose an aggregate ranking-based metric that equally weights precision and recall, as well as equally weights the various evaluation metrics. Please see Appendix C for details.

---

> > ### Comment · Reviewer_vpJf · 2022-11-21
> > **Response**
> >
> > I would like to thank the authors for their time taken to respond and adapt their paper to my comments. After reading the response, where the authors resolve some issues, i still have doubts about the significance and novelty of the work. This venue of work requires more systematic, more inclusive, more extended treatment. This work is only a good first step towards that. My scores remain unchanged.

---

> > > ### Author Response · Authors · 2022-11-22
> > > **Authors Response**
> > >
> > > Thank you for your prompt response. We very much appreciate the consideration and your comments, and we would kindly ask for further elaboration on the points that remain open.
> > >
> > > To summarize, you requested:
> > > - Inclusion of CHIP-seq derived networks (Pratapa et al. 2020) - which we have added (see Fig 2 in the revised manuscript).
> > > - Inclusion of classic methods for GRN inference, such as SCENIC (Aibar et al 2017) which we have added (see Fig 2 and p. 5 Baselines in the revised manuscript).
> > > - A consensus score that aggregates the various metrics into a single ranking of methods to improve interpretation - which we have added (see Appendix C Tables C1 and C2 in the revised manuscript).
> > > - Implementation of a quality control procedure for the perturb-Seq datasets - which we have added via a perturbation as well as cell-level quality control filtering following Replogle et al 2022 (please see description of the procedure in our response above).
> > > - Documenting the partition sizes used for each method to enable running them on large gene-gene graphs - which we added (please see Appendix B Table B1 in the revised manuscript).
> > > - Clarification whether the use of the word “significant” on p.9 in the submitted manuscript in relation to Fig. 3 corresponded to a statistical test - to which we responded that the use of the word “significant” was not referring to statistical significance in that paragraph (see our response above). We are nonetheless happy to run additional random seeds to enable testing for statistical significance to further substantiate Fig. 3 in the final version of the manuscript (we have not originally due to the large computational requirements).
> > >
> > > We intended for the above amendments to fully resolve your comments raised, and would therefore appreciate further elaboration as to which points were not yet resolved. We will be happy to make any further amendments necessary to resolve your comments.
> > >
> > > Reviewer vpJf: “i still have doubts about the significance and novelty of the work. This venue of work requires more systematic, more inclusive, more extended treatment.”
> > >
> > > We thank Reviewer vpJf for this additional comment. Given the original review recognizing the empirical novelty of our work, we would appreciate it if you could detail what concrete improvements the Reviewer believes could be made to the manuscript to make it “more systematic, more inclusive, more extended”.
> > > - Significance and novelty: As shown by prior benchmarking efforts (see e.g. ImageNet in computer vision), open benchmarks can considerably accelerate progress towards better methods, are crucial to measure and stimulate further method development and are hence of paramount significance for machine learning research - even if they do not contribute to a narrow definition of technical novelty. In CausalBench, we - for example - clearly delineate a number of open challenges that could be addressed with improved machine learning methods for integrating perturbation information from single-cell experiments for GRN inference that were not known prior. In addition, CausalBench is the first benchmark for causal inference at scale that includes a large number of perturbational data points in a setting with gene-gene networks of realistic size (1’000+ nodes).
> > > - Inclusive: Moreover, it is difficult for machine learning researchers without domain knowledge to contribute solutions to important biological solutions without openly available and easily accessible benchmarks - we therefore believe inclusivity is one of the strengths of our work.
> > > - Systematic: CausalBench presents a systematic and unbiased methodology for comparing the outputs to both biological domain knowledge as well as statistical ground truth that is suitable for evaluating a wide range of methods - as demonstrated by the variety of methods evaluated in our experiments covering classical GRN methods, causal inference methods and state-of-the-art baselines.
> > > - Extensive: To the best of our knowledge, with (i) two of the largest dataset evaluated to date both in number of cells and perturbations in two cell types, (ii) with the largest gene-gene interaction networks (1’000 nodes) covered by any benchmark to date, (iii) with 11 baseline methods evaluated, (iv) 7 different ground truth evaluations conducted and (v) a sensitivity analysis to number of samples and interventions our benchmark is the most extensive benchmark of GRN inference from perturbed scRNAseq data to date - and we therefore consider this a strength of our proposed benchmark relative to existing works.
> > >
> > > We would be happy to receive your clarifications. Many thanks in advance.

---

> > > > ### Comment · Reviewer_vpJf · 2022-11-23
> > > > **Response**
> > > >
> > > > The authors have addressed the comments, but still there are some crucial points that are missing. While it is true that this work is the first one to use single-cell perturbation experiments, the only possible advantage of using such resolution is an increased number of samples to train on, even though the authors admit that these are not true replicates, and other benchmarks have used perturbation experiments for evaluation of GRNs (at a better gene resolution since they use bulk transcriptomics).
> > > >
> > > > The benchmark is also not systematic enough, the fact that adding one of the most commonly used non-causal methods performs the best in a causal benchmark is concerning. This is evident even without measures of significance of differences in performance. It however raises the questions about the significance of the work at this particular moment. It also raises the question on whether the benchmarked approaches are suitable for single-cell data. Also, this one particular method (SCENIC) is only one of many existing methods. Given the results, it is surprising that the authors didn't ask the question whether other well-known non-causal method beyond the one explicitly mentioned will outperform the causal methods and why is that so.
> > > >
> > > > The addition of biological and predictive metrics is a good step in the right direction, however they are still quite limited. There is no real contextualized, but only "general" ground truth available (e.g. STRING), which again raises the question why single cell for causal methods.
> > > >
> > > > I really do appreciate the effort made by the authors and I believe that addressing the comments in the way they were addressed somewhat improved their paper. I also read and considered the important issues raised by other reviewers. I will increase the score of my review, but remain at the threshold.

---

> ### Author Response · Authors · 2022-11-19
> **Authors Response (Part 2) to Official Review of Paper5185 by Reviewer vpJf**
>
> C4: Although the authors are aware of the limitation of perturbation experiments, no quality control assessment is performed to test whether the perturbation actually worked. Samples where the perturbed gene still shows high levels of gene expression should be removed or at least accounted for.
>
> AR: We appreciate your suggestion to include only those perturbations that actually result in a measurable knockdown effect.
> We performed this quality control at two levels. The first is at the level of perturbations and the second is at the level of individual cells. At the level of perturbations, we followed the same criteria as used in Replogle et al. 2022. Strong perturbations were identified as those that satisfied the following conditions: (i) at least 50 differentially expressed genes at a significance of p < 0.05 by Anderson-Darling test following Benjamini-Hochberg correction; (ii) at least 25 cells that passed our quality filters; and (3) an on-target knockdown, if measured, of at least 30%. This reduced the number of perturbations in the K562 cell line from 2058 to 1092 and in RPE-1 from 2394 to 1543. The number of cells was reduced from 310385 to 192648 for K562 and from 247914 to 175398 for RPE-1.
>
> At the level of individual cells, we tested whether each perturbation was achieving its intended effect by comparing the gene expression level of the perturbed gene (say X) following perturbation to its original level before perturbation, as you recommended. We set as a threshold the expression level of gene X in the tenth percentile cell in the unperturbed control distribution. Any cell where X is perturbed but the expression of X is over this threshold was not considered. It is also important to note that the expression of a perturbed gene is not always measured in the dataset, in which case no filtering was performed. As a consequence of these filterings, the number of cells was reduced from 192648 to 162751 for K562 and from 175398 to 162733 for RPE-1.
>
> We nevertheless keep this filtering as an option in our benchmark suite which can be improved or altered easily, as future users may come up with better filtering methods or take advantage of the additional data.
>
>
> C5: The partition of the data due to scalability issues raises concerns since, as the authors already mention, it breaks the no-latent-confounder assumption. Authors chose partition sizes such that the running time remains below 30 hours but do not mention anywhere the actual numbers for each method. In order to comply with this rule, some methods might use fractions of the feature space that are too small to generate valuable graphs.
>
> AR: This information was added in Appendix B of the revised manuscript.
>
> C6: In figure 3, authors mention “significant” increase but do not perform any statistical test to corroborate it.
>
> AR: Every experiment was run for 3 random seeds (as our computing resources allowed). In Figure 3, for every method, there are three points of the same color at each dataset ratio,  corresponding to the three runs. As can be seen, in most cases, the intra-method variance due to random seed is small compared to inter-method differences which indicates the robustness of the analysis to randomness.

---

### Official Review · Reviewer_WpyP · 2022-10-24

**Confidence:** 4
**Correctness:** 3
**Technical Novelty And Significance:** 2
**Empirical Novelty And Significance:** 2
**Recommendation:** 5

**Clarity, Quality, Novelty And Reproducibility:**

- The authors claim to have all code available in a github repository, but the repository seems to be empty.
- Overall, I think the paper lacks technical precision needed for publication.  There are many cases where wordy but vague descriptions are used in place of more precise statements.  E.g. "satisfactory results are only obtained on the RPE1 dataset".  What counts as satisfactory?  What is the reason the other results are unsatisfactory? At the same there is additional and unneeded notation and technical details regarding SCM.  In my opinion, this doesn't add much, since you are not describing a new SCM-based method and it is never referenced again.  Thus the background on SCMs seems superflous.
- Re-check for typos and language, e.g. "appraoch"


**Strength And Weaknesses:**

- I think the paper tackles an important problem and provides a great service by colating these datasets and metrics.
- My main complaint is that the paper advertises it's contribution as providing a benchmark for network inference on large scale scRNA seq data although they admit that there are no gold standard datasets available from which to actually benchmark these methods.  This is a noted limitation, but my concern is that this is a really big limitation.  It's really hard to know whether the results are biologically meaningful, or really just noisy and reflect the difficulty of identifying useful benchmarks.
- Along these lines, I think the "Metrics" section needs a lot more clarification.  The authors refer to equality tests on "the two distributions" as well as Wasserstein distance on "the two empirical distributions".  What are these distributions of? There could be a lot more detail here regarding the data / statistics used to create these distributions and how the tests were calculated.
- While I understand some of the difficulties, it's hard to know how much to read into the TPR vs TP counts graphs.  The authors need to explicitly discuss the potential for false positives / false positive rates.  Are there any ways to identify putative false positives that can be used as a pseudo-benchmark?



**Summary Of The Paper:**

The authors of this paper propose a suite of tools for benchmarking network inference on both observational and interventional data single-cell gene expression data.  They run several network inference / causal discovery methods for recovering graphical relationships from scRNAseq data and evaluate the performance of the methods using existing putatative knowledge about specific gene-gene interactions as well as by comparing performance of observational and interventional data.

**Summary Of The Review:**

Overall, I think the authors take on a challenging and important task of collating multiple datasets and implementing many causal discovery methods on these datasets.  Overall, I think the paper lacks clarity and justification around the statistical benchmarks employed.  As such, it's unclear how much value is actually added in the suite of benchmarks proposed.

---

> ### Author Response · Authors · 2022-11-19
> **Authors Response (Part 1) to Official Review of Paper5185 by Reviewer WpyP**
>
> Thank you for recognizing the importance of this work as a step towards benchmarking biological networks and also pointing out the limitations of this type of work. We address each of your concerns below. In case an answer requires a change in the manuscript, the change is colored red in the revised manuscript for better visibility
> (Cx: Reviewer’s Concern number x, AR: Author Response)
>
>
> C1: My main complaint is that the paper advertises it's contribution as providing a benchmark for network inference on large scale scRNA seq data although they admit that there are no gold standard datasets available from which to actually benchmark these methods. This is a noted limitation, but my concern is that this is a really big limitation. It's really hard to know whether the results are biologically meaningful, or really just noisy and reflect the difficulty of identifying useful benchmarks.
>
> AR: We thank the reviewer for their important comments on gold-standard datasets. Gene regulatory networks have been studied extensively within the bioinformatics and computer science communities for over two decades. One of the first seminal papers looking at this problem inferred these networks using linear regression over microarray data (Friedman et al. 2000). Since then, despite a greater dearth of experimental validation data in the past than today, the algorithmic advances in such network inference approaches have led to meaningful progress in the identification and interpretation of biological processes (Fleck et al. Nature 2022,  Wagner et al. Nature Biotechnology 2016, Aibar et al. Nature Methods 2017, Pratapa et al. Nature Methods 2020). Thus, while the inference of true causal graph structure is appealing to the causal inference community, the biological community has been able to make significant scientific advances with the data that is available to them.
>
> Moreover, we believe that one of the key contributions of our work is the use of a combination of orthogonal metrics across different cell types and datasets. This allows us to come closer to approximating ground truth than has been possible in the past where most analyses were focused on just a single metric. We are also the first benchmark to leverage perturbation data, which allows us to measure performance (through the statistical evaluation metric) against a larger space of putative gene regulatory interactions than would be possible using biological databases alone.
>
> Causalbench is a forward-looking benchmark since we are providing a framework for evaluating network inference procedures using large-scale gene perturbation experiments which are likely to become more and more ubiquitous over time and are the most economically viable strategy for inferring regulatory relationships at scale. Thus, by providing a comprehensive evaluation of algorithmic performance over the rapidly growing single-cell perturbational data, we expect CausalBench to become an increasingly important resource for both the biological and causal inference communities.
>
> C2: Along these lines, I think the "Metrics" section needs a lot more clarification. The authors refer to equality tests on "the two distributions" as well as Wasserstein distance on "the two empirical distributions". What are these distributions of? There could be a lot more detail here regarding the data / statistics used to create these distributions and how the tests were calculated.
>
> AR: We based our analysis on two metrics, biologically-inspired metrics that cross-check the discovered graph with the existing biological knowledge and a statistical metric that only uses the existing data to cross-validate the discovered edges in the GRN. The distributional equality test and Wassertesin distance belong to the latter class of metrics that are built upon this assumption: In the GRN network, if there is a directed link or a path from gene A to gene B, an intervention on Gene A with high probability will change the distribution of values that gene B takes in the transcription profile. The “two distributions” refers to the distribution of values that B takes on under the observational setting (where no intervention on A is performed) and the interventional setting (where A is perturbed). We thank the reviewer for their comments and have updated the language in the text to further clarify this (please see the paragraph on Quantitative statistical evaluation in Section 3).

---

> ### Author Response · Authors · 2022-11-19
> **Authors Response (Part 2) to Official Review of Paper5185 by Reviewer WpyP**
>
> C3: While I understand some of the difficulties, it's hard to know how much to read into the TPR vs TP counts graphs. The authors need to explicitly discuss the potential for false positives / false positive rates. Are there any ways to identify putative false positives that can be used as a pseudo-benchmark?
>
> AR: We thank the reviewer for their comments and would like to acknowledge an error in the original plots which show precision v/s TP and not TPR v/s TP. However, in the revised manuscript, we will be updating these plots to now show precision v/s recall with the caveat that we only have an upper bound on false positives.
>
>
>
> C4: The authors claim to have all code available in a github repository, but the repository seems to be empty.
>
> AR: We apologize for an error that unintentionally led to the shared repository being empty. We regret the delay this caused in granting access to the code. The source code is now available, as well as an appendix explaining how to use the benchmark.
>
>
>
> C5: Overall, I think the paper lacks technical precision needed for publication. There are many cases where wordy but vague descriptions are used in place of more precise statements. E.g. "satisfactory results are only obtained on the RPE1 dataset". What counts as satisfactory? What is the reason the other results are unsatisfactory? At the same there is additional and unneeded notation and technical details regarding SCM. In my opinion, this doesn't add much, since you are not describing a new SCM-based method and it is never referenced again. Thus the background on SCMs seems superflous. Re-check for typos and language, e.g. "appraoch"
>
>
> AR: We thank the reviewer for their comments. We have incorporated this feedback and added more precise language to describe the metrics in the updated manuscript (please see the paragraph on Quantitative statistical evaluation in Section 3) . Our proposed aggregate metric offers a more principled way to compare models based on all our metrics (please see Appendix C).
>
> We agree that the description of the SCM framework was not well motivated in the first submitted version of the manuscript and in particular not well connected to the storyline as alluded to by the reviewer. To amend this, we have better connected the SCM foundations with the rest of the paper by motivating its use and necessity for describing the setting in a causal language and setting the scene for the later on described methods addressing this setting.

---

> ### Author Response · Authors · 2022-11-30
> **Authors message to Reviewer WpyP**
>
> Dear Reviewer WpyP,
>
> We thank you again for your feedback and suggestions to improve our work. We would like to kindly ask you if you could reevaluate your review in light of the revised manuscript and the rebuttal discussion. We thank you in advance for your time and consideration, and we would be happy to further discuss open points with you.
>
> Kind regards,
>
> The Authors

---

> ### Comment · Reviewer_WpyP · 2022-12-01
> **Increased score to 5**
>
>  I commend the authors for their extensive response to my review.  I have increased my score to 5 but remain below the acceptance threshold. However, I remain somewhat skeptical about the methodological novelty and contributions in this work.  While the data might end up  being a useful resource for the biological communities, but I don't think the proposed evaluation metrics are a significant enough contribution.

---

### Official Review · Reviewer_QjTd · 2022-10-24

**Confidence:** 4
**Clarity, Quality, Novelty And Reproducibility:** The paper lacks clarity and novelty.
**Correctness:** 2
**Technical Novelty And Significance:** 1
**Empirical Novelty And Significance:** 1
**Recommendation:** 3

**Strength And Weaknesses:**

The paper is hard to follow for me. From the abstract and introduction, it seems it aims at providing a dataset / benchmark for GRN inference that would be very easy to understand and use by researchers from outside of computational biology or network inference fields - something like DREAM 3/4/5 GRN challenges from a decade ago. However, starting from Section 3, the authors introduce a very specific formal structure (SCM) that also does not necessarily reflect the underlying biology (e.g. the DAG assumption) - it is not clear what purpose this serves. The choice of benchmark evaluation metrics (Section 4) seems somewhat ad-hoc (why STRING and not other network repository, why use just true positives instead of metrics based on precision-recall and ROC used in DREAM). On the other hand, the paper lacks details about how to actually use the dataset in practice, whether there is a clear leaderboard etc. - details that would be essential in establishing CausalBench as the benchmark for the field. It does not help that the github repository mentioned in the introduction is empty.

**Summary Of The Paper:**

The paper aims to establish a dataset / benchmark for gene regulatory network inference from single-cell data.

**Summary Of The Review:**

The paper fails in achieving what a good benchmark/dataset paper should demonstrate: clarity, sound assumptions and design choices, and practicality. Reading it did not bring me any closer to being able to quickly, effortlessly evaluate a GRN inference method I may develop.

---

> ### Author Response · Authors · 2022-11-18
> **Authors Response (Part1) to Official Review of Paper5185 by Reviewer QjTd**
>
> Thank you for your feedback on our work. We address each of your concerns below.
> (Cx: Reviewer’s Concern number x, AR: Author Response)
>
>
> C1: The paper is hard to follow for me. (...) the authors introduce a very specific formal structure (SCM) that also does not necessarily reflect the underlying biology (e.g. the DAG assumption) - it is not clear what purpose this serves.
>
> AR: We appreciate the reviewer’s comments and we have improved in the revised manuscript version that we are posting shortly.  We agree that the description of the SCM framework was not well motivated in the first submitted version of the manuscript and in particular not well connected to the storyline as alluded to by the reviewer. To amend this, we have better connected the SCM foundations with the rest of the paper by motivating its use and necessity for describing the setting in a causal language and setting the scene for the later on described methods addressing this setting.

---

> ### Author Response · Authors · 2022-11-18
> **Authors Response (Part2) to Official Review of Paper5185 by Reviewer QjTd**
>
> C2: The choice of benchmark evaluation metrics (Section 4) seems somewhat ad-hoc (why STRING and not other network repository, why use just true positives instead of metrics based on precision-recall and ROC used in DREAM).
>
>
> AR: We agree that the choice of biological repository is paramount to evaluate the right qualities of interest for network inference algorithms – in order to broadly cover multiple known biological interaction types we included several repositories containing known relationships from different angles: the STRING and CORUM database covering known protein-protein interactions (PPI) and protein complexes, respectively. In addition, in the revised manuscript, we have added CHIP-Seq networks to further increase our coverage of ground truth biological repositories.
>
> The three data repositories that we included in the benchmark represent some of the highest quality, genome-wide functional interaction networks that are openly available today: STRING [1]  is one of the most commonly used PPI network reference databases that is continuously updated with the latest information from biological assays as well as literature and commonly used as a ground truth dataset in interventional experiment analyses. The CORUM database represents some of the most well cartographed functional gene interactions in their formation of protein complexes that have been studied across many investigations [2]. Lastly, the CHIP-Seq dataset that we introduced in the revised manuscript was previously used in a number of studies to evaluate functional gene interactions [3] and is hence an accepted standard for evaluating network inference algorithms in the community. Please kindly let us know if there are any other high quality, genome-wide databases that we could potentially include as a ground truth data source - in general, while many biological repositories are available in literature, for the purposes of our benchmark, we need repositories that cover interactions on a genome-wide scale to not introduce pathway biases.
>
> A key contribution of our work is the use of a combination of orthogonal metrics and benchmark datasets across different cell types and settings to evaluate algorithms. This allows us to better approximate ground truth than has been possible with previous analyses that were focused on just a single metric. We are also the first benchmark to leverage perturbation data. This allows us to measure performance (through the statistical evaluation metric) against a larger space of putative gene regulatory interactions than would be possible using biological databases alone. Moreover, network inference methods trained on interventional data should theoretically come closer to estimating the true underlying causal graph since interventional data is better able to restrict the size of the Markov equivalence classes as compared to existing benchmarks based on observational data alone [3].
>
> In addition to this, we propose a combined score as per your suggestion, which also allows us to create a ranked scoreboard of algorithms. Details for this new composite score can be found in the appendix. To address your comment regarding metrics used, we have updated the revised manuscript to include a visualization with the precision-recall metric as previously used in DREAM. While we are aware that this evaluation approach has been used in related works, we felt that the assumptions to compute such metrics did not hold as strongly in our case (for example, given that the biological networks only contain partial ground-truth, the computed precision is only a lower-bound to the “true” precision score of an output). We will move the plots that highlight the size of the output graph to the Appendix and also outline some of the limitations of relying on precision-recall in this setting to the revised manuscript.
>
>
> [1] Damian Szklarczyk, Annika L Gable, Katerina C Nastou, David Lyon, Rebecca Kirsch, Sampo Pyysalo, Nadezhda T Doncheva, Marc Legeay, Tao Fang, Peer Bork, et al. The string database in 2021: customizable protein–protein networks, and functional characterization of user-uploaded gene/measurement sets. Nucleic acids research, 49(D1):D605–D612, 2021
>
> [2] Madalina Giurgiu, Julian Reinhard, Barbara Brauner, Irmtraud Dunger-Kaltenbach, Gisela Fobo, Goar Frishman, Corinna Montrone, and Andreas Ruepp. Corum: the comprehensive resource of mammalian protein complexes—2019. Nucleic acids research, 47(D1):D559–D563, 2019.
>
> [3] Aditya Pratapa, Amogh P Jalihal, Jeffrey N Law, Aditya Bharadwaj, and TM Murali. Benchmarking algorithms for gene regulatory network inference from single-cell transcriptomic data. Nature methods, 17(2):147–154, 2020.

---

> ### Author Response · Authors · 2022-11-18
> **Authors Response (Part3) to Official Review of Paper5185 by Reviewer QjTd**
>
> C3: On the other hand, the paper lacks details about how to actually use the dataset in practice, whether there is a clear leaderboard etc. - details that would be essential in establishing CausalBench as the benchmark for the field. It does not help that the github repository mentioned in the introduction is empty.Reading [the paper] did not bring me any closer to being able to quickly, effortlessly evaluate a GRN inference method I may develop.
>
>
> AR: We apologize for an error that unintentionally led to the shared repository being empty. We regret the delay this caused in obtaining access to the code. The source code is now openly available at the indicated URL (https://github.com/ananymous-43213123/causalbench), and we would encourage the reviewer to kindly revisit the repository with an included README file that outlines the usage of the benchmark with new methods. For reader’s convenience and by the kind feedback of yours, we have additionally added a new appendix to the revised manuscript that outlines how to use the benchmark with new methods. We hope that this concern is now addressed and we would greatly appreciate it if you would reevaluate your feedback in light of this additional information.
>
>
>
> C4: it seems it aims at providing a dataset / benchmark for GRN inference that would be very easy to understand and use by researchers from outside of computational biology or network inference fields - something like DREAM 3/4/5 GRN challenges from a decade ago.
>
>
> CausalBench is a gene regulatory network inference benchmark providing a framework for evaluating network inference procedures using new technologies such as perturb-Seq, that for the first time generate large-scale gene experiment data under perturbations which are likely to become more and more ubiquitous over time as this technology is widely adopted. Thus, by providing a comprehensive evaluation of algorithmic performance over the rapidly growing single-cell perturbational data, we hope for CausalBench to become an increasingly important resource for both the biological and causal inference communities.

---

> ### Comment · Reviewer_QjTd · 2022-11-22
> **updated score**
>
> I appreciate authors' efforts to improve the manuscript, and raise my score from 1 to 3.

---

> > ### Author Response · Authors · 2022-11-25
> > **Authors response**
> >
> > We thank you for your updated score and for recognizing our efforts in improving the manuscript. Our understanding was that your main concerns resided around the practicability of using our benchmark as well as its assumptions and design. We intended for our amendments and additional information around the benchmark design and usage to fully resolve your comments raised, and would therefore appreciate further elaboration as to which points are not yet resolved, or further points that you see could be improved. We thank you in advance for your continued consideration.

---

### Author Response · Authors · 2022-11-19
**Authors Response to all Reviewers**

Dear Reviewers,

We would like to thank you for engaging with our manuscript and are really grateful for the thoughtful comments. In response to your reviews, we have made several changes to the text and figures, implemented a new baseline, and conducted additional experiments using an improved preprocessing pipeline. We believe the submission has improved significantly as a result. Here is the summary of the key feedback points and how we addressed them:
- Cell-specific evaluation: we added a biological evaluation based on cell-specific CHiP-seq networks (Reviewer vpJf and Reviewer 2A85)
- Leaderboard: we proposed a simple and unbiased way to combine the diverse metrics we implemented (Reviewer QjTd and Reviewer vpJf)
- Access to the source code: the code is now publicly available for review (Reviewer WpyP and Reviewer QjTd)
- Clarification around the necessity of the causal motivation of the setting and of the quantitative metrics (Reviewer QjTd, Reviewer 2A85, and Reviewer WpyP)

We are answering all reviewer comments in detail below. We would be so glad to engage further during the rest of the discussion period to resolve any remaining concerns or questions about the work.

Kind regards,

The authors

## ERRATUM :
We realised that we sent an outdated version of the new draft. We would like to humbly apologise for this mistake and would like to kindly ask the reviewers to consider the updated manuscript under this link: https://github.com/ananymous-43213123/causalbench/blob/main/Causalscbench_ICLR2023.pdf. We apologise again for the inconvenience.

---

### Decision · Program_Chairs · 2023-01-20

**Decision:**

Reject

**Justification For Why Not Higher Score:**

Most reviewers recommended rejecting the paper.

**Justification For Why Not Lower Score:**

N/A

**Metareview: Summary, Strengths And Weaknesses:**

  The authors propose a new benchmark for regulatory network inference/causal discovery based on single cell RNAseq data with interventional perturbations. While such a benchmark would be valuable for the community, the reviewers identified several issues with the authors' proposal and have recommended against accepting the paper. Two major issues were the benchmark does indeed reflect the underlying biology, and whether the proposed evaluation metrics are appropriate.